# Quantification of fibroblast growth factor 23 and N-terminal pro-B-type natriuretic peptide to identify patients with atrial fibrillation using a high-throughput platform: A validation study

Winnie Chua[1], Jonathan P. Law[1,2], Victor R. Cardoso[1,3], Yanish Purmah[1,4,5], Georgiana Neculau[4,5], Muhammad Jawad-Ul-Qamar[1,4,5], Kalisha Russell[5], Ashley Turner[5], Samantha P. Tull[1], Frantisek Nehaj[1,5], Paul Brady[1,4,5], Peter Kastner[6], André Ziegler[7], Georgios V. Gkoutos[3], Davor Pavlovic[1], Charles J. Ferro[1,2], Paulus Kirchhof[1,4,5,8,9], Larissa Fabritz[1,4,8]*

1 Institute of Cardiovascular Sciences, University of Birmingham, Birmingham, United Kingdom, 2 Department of Renal Medicine, University Hospitals Birmingham NHS Foundation Trust, Birmingham, United Kingdom, 3 Institute of Cancer and Genomic Sciences, University of Birmingham, Birmingham, United Kingdom, 4 Department of Cardiology, University Hospitals Birmingham NHS Foundation Trust, Birmingham, United Kingdom, 5 Sandwell and West Birmingham NHS Trust, Birmingham, United Kingdom, 6 Roche Diagnostics GmbH, Penzberg, Germany, 7 Roche Diagnostics International AG, Rotkreuz, Switzerland, 8 University Heart and Vascular Center UKE Hamburg, Hamburg, Germany, 9 German Center for Cardiovascular Research (DZHK), partner site Hamburg/Kiel/Lübeck, Germany

* L.Fabritz@bham.ac.uk

## Abstract

### Background

Large-scale screening for atrial fibrillation (AF) requires reliable methods to identify at-risk populations. Using an experimental semi-quantitative biomarker assay, B-type natriuretic peptide (BNP) and fibroblast growth factor 23 (FGF23) were recently identified as the most suitable biomarkers for detecting AF in combination with simple morphometric parameters (age, sex, and body mass index [BMI]). In this study, we validated the AF model using standardised, high-throughput, high-sensitivity biomarker assays.

### Methods and findings

For this study, 1,625 consecutive patients with either (1) diagnosed AF or (2) sinus rhythm with $CHA_2DS_2$-VASc score of 2 or more were recruited from a large teaching hospital in Birmingham, West Midlands, UK, between September 2014 and February 2018. Seven-day ambulatory ECG monitoring excluded silent AF. Patients with tachyarrhythmias apart from AF and incomplete cases were excluded. AF was diagnosed according to current clinical guidelines and confirmed by ECG. We developed a high-throughput, high-sensitivity assay for FGF23, quantified plasma N-terminal pro-B-type natriuretic peptide (NT-proBNP) and FGF23, and compared results to the previously used multibiomarker research assay. Data were fitted to the previously derived model, adjusting for differences in measurement

**Data Availability Statement:** The data generated and analysed in this study cannot be shared

publicly because of patient consent and ethics approval restrictions. Reasonable requests / collaborations to use the data will be considered by contacting Research Governance at the University of Birmingham, UK (researchgovernance@contacts.bham.ac.uk).

**Funding:** This work was partially supported by the European Commission (grant agreements no. 633196 [CATCH ME]) and no. 116074 [BigData@Heart EU IMI] to PKi and LF, British Heart Foundation (FS/13/43/30324 to PKi and LF), German Centre for Cardiovascular Research supported by the German Ministry of Education and Research (DZHK, via a grant to AFNET to PKi), and Leducq Foundation to PKi. The Institute of Cardiovascular Sciences has received the British Heart Foundation (BHF) Accelerator Award (AA/18/2/34218). https://www.bhf.org.uk/for-professionals/information-for-researchers/managing-your-grant/open-access-policy https://ec.europa.eu/programmes/horizon2020/en http://www.ehnheart.org/projects/imi-bigdata-heart/1086:imi-bigdataheart.html https://www.kompetenznetz-vorhofflimmern.de/en/research/dzhk-studies The funders had no role in study design, data collection and analysis, decision to publish, or preparation of the manuscript.

**Competing interests:** I have read the journal's policy and the authors of this manuscript have the following competing interests: LF has received institutional research grants and non-financial support from European Union, British Heart Foundation, Medical Research Council (UK), and DFG and Gilead. PK and LF are listed as inventors on two patents held by University of Birmingham (Atrial Fibrillation Therapy WO 2015140571, Markers for Atrial Fibrillation WO 2016012783). PKi has received additional support for research from the European Union, British Heart Foundation, Leducq Foundation, Medical Research Council (UK), and German Centre for Heart Research, from several drug and device companies active in atrial fibrillation, honoraria from several such companies. PKa is an employee of Roche Diagnostics GmbH. AZ is an employee of Roche Diagnostics Intl. All other authors have reported no relationships relevant to the contents of this paper to disclose.

**Abbreviations:** AF, atrial fibrillation; AUC, area under the receiver operating characteristic curve; BBC-AF Registry, Birmingham-Black Country-Atrial Fibrillation registry; BMI, body mass index; BNP, B-type natriuretic peptide; CKD, chronic kidney disease; CKD-EPI, Chronic Kidney Disease Epidemiology Collaboration; eGFR, estimated glomerular filtration rate; FGF23, fibroblast growth factor 23; MDRD, Modification of Diet in Renal

platforms and known confounders (heart failure and chronic kidney disease). In 1,084 patients (46% with AF; median [Q1, Q3] age 70 [60, 78] years, median [Q1, Q3] BMI 28.8 [25.1, 32.8] kg/m$^2$, 59% males), patients with AF had higher concentrations of NT-proBNP (median [Q1, Q3] per 100 pg/ml: with AF 12.00 [4.19, 30.15], without AF 4.25 [1.17, 15.70]; $p < 0.001$) and FGF23 (median [Q1, Q3] per 100 pg/ml: with AF 1.93 [1.30, 4.16], without AF 1.55 [1.04, 2.62]; $p < 0.001$). Univariate associations remained after adjusting for heart failure and estimated glomerular filtration rate, known confounders of NT-proBNP and FGF23. The fitted model yielded a C-statistic of 0.688 (95% CI 0.656, 0.719), almost identical to that of the derived model (C-statistic 0.691; 95% CI 0.638, 0.744). The key limitation is that this validation was performed in a cohort that is very similar demographically to the one used in model development, calling for further external validation.

## Conclusions

Age, sex, and BMI combined with elevated NT-proBNP and elevated FGF23, quantified on a high-throughput platform, reliably identify patients with AF.

## Trial registration

Registry IRAS ID 97753 Health Research Authority (HRA), United Kingdom

## Author summary

### Why was this study done?

- Atrial fibrillation (AF) is the most commonly sustained arrhythmia in the world, markedly increasing the risk of stroke, heart failure, and cardiovascular death.

- AF episodes can be transient and go undetected by ECG for a long time, leading to untreated AF and avoidable complications such as stroke.

- Blood biomarkers are an easy and affordable way to enable screening for undiagnosed, silent AF.

- An earlier study using a dual-antibody-based research assay identified a simple model, consisting of age, sex, body mass index, and 2 biomarkers, FGF23 and NT-proBNP, that identified patients with AF.

### What did the researchers do and find?

- We measured 2 biomarkers, NT-proBNP and FGF23, using high-throughput, high-sensitivity assays that are ready for clinical use. The FGF23 assay was developed for this study.

- We incorporated these biomarkers into a model including age, sex, and body mass index, which was derived from previous work where research assays were used to identify biomarkers to detect patients with AF.

Disease; NT-proBNP, N-terminal pro-B-type natriuretic peptide; ROC, receiver operating characteristic.

- We demonstrated that the model using the standardised assays performed consistently to identify patients with AF.

### What do these findings mean?

- Elevated concentrations of FGF23 and NT-proBNP, in addition to older age, female sex, and high body mass index, identify patients who are likely to have AF.

- In the future, point-of-care testing can be easily performed using these biomarkers in combination with 3 simple pieces of information (age, sex, and body mass index) to target intensive screening, for example, using ECG monitoring.

## Introduction

Improving detection of patients with atrial fibrillation (AF) has the potential to prevent stroke, premature cardiovascular death, and dementia [1–3], but large-scale ECG screening of entire populations is difficult to justify, as stated, for example, by the US Food and Drug Administration (FDA). Targeting ECG screening to populations at risk could render AF screening feasible and cost-effective [4,5]. B-type natriuretic peptide (BNP) and fibroblast growth factor 23 (FGF23) were previously identified from an initial pool of 92 biomarkers to be strongly associated with prevalent AF in a cohort of consecutive patients presenting to hospital [6]. These biomarkers were discovered using both statistical modelling and machine learning techniques that incorporated processes for biomarker selection and considered key clinical and imaging parameters alongside biomarkers. We found that the combination of age, sex, and body mass index (BMI) with BNP and FGF23 was able to discriminate patients with AF from patients in sinus rhythm [6]. In that study, biomarker concentrations were estimated using a research platform with arbitrary biomarker units.

In this current study, we developed a novel high-throughput platform for quantifying FGF23, used a clinically utilised platform for N-terminal pro-B-type natriuretic peptide (NT-proBNP), and quantified these biomarkers in the previously tested patients, as well as in an unselected independent cohort of patients presenting to hospital, for validation. To assess the predictive power of these biomarkers combined with simple metrics (age, sex, and BMI), we carefully corrected our associations for known confounders of elevated FGF23 and natriuretic peptides. As AF screening is aimed at identifying modifiable risks of stroke and other cardiovascular complications, we focused our study on patients with $CHA_2DS_2$-VASc score of 2 or more as there is a clear indication for initiation of therapy (e.g., anticoagulation therapy) in this group of patients.

## Methods

### Study population

Consecutive patients referred to the Sandwell and West Birmingham Sandwell and West Birmingham NHS Trust (Birmingham, UK) NHS Trust (Birmingham, UK) for inpatient or outpatient evaluation of acute illnesses were recruited between September 2014 and February 2018 into the Birmingham and Black Country Atrial Fibrillation Registry (BBC-AF Registry). Patients eligible for recruitment either had diagnosed AF confirmed by ECG, or $CHA_2DS_2$-VASc score of 2 or more. All patients without diagnosed AF underwent 7-day ambulatory

ECG monitoring to exclude silent AF, as previously published [6]. All patients were phenotyped: Clinical information was collected during a detailed interview at time of blood sampling, review of electronic patient records, and clinical chart review. ECG and transthoracic echocardiography were performed in all patients.

In the primary analysis, patients with tachyarrhythmias apart from AF (atrial high rate episodes, $n = 3$; atrial flutter, $n = 34$) were excluded. Sensitivity analyses were performed including patients with atrial flutter. Using complete cases, we evaluated the performance of the model in patients whose data had not been used to derive the original model (Fig 1).

### Ethics statement

This study complied with the Declaration of Helsinki, was approved by the National Research Ethics Service Committee (BBC-AF Registry, West Midlands, UK, IRAS ID 97753), and was

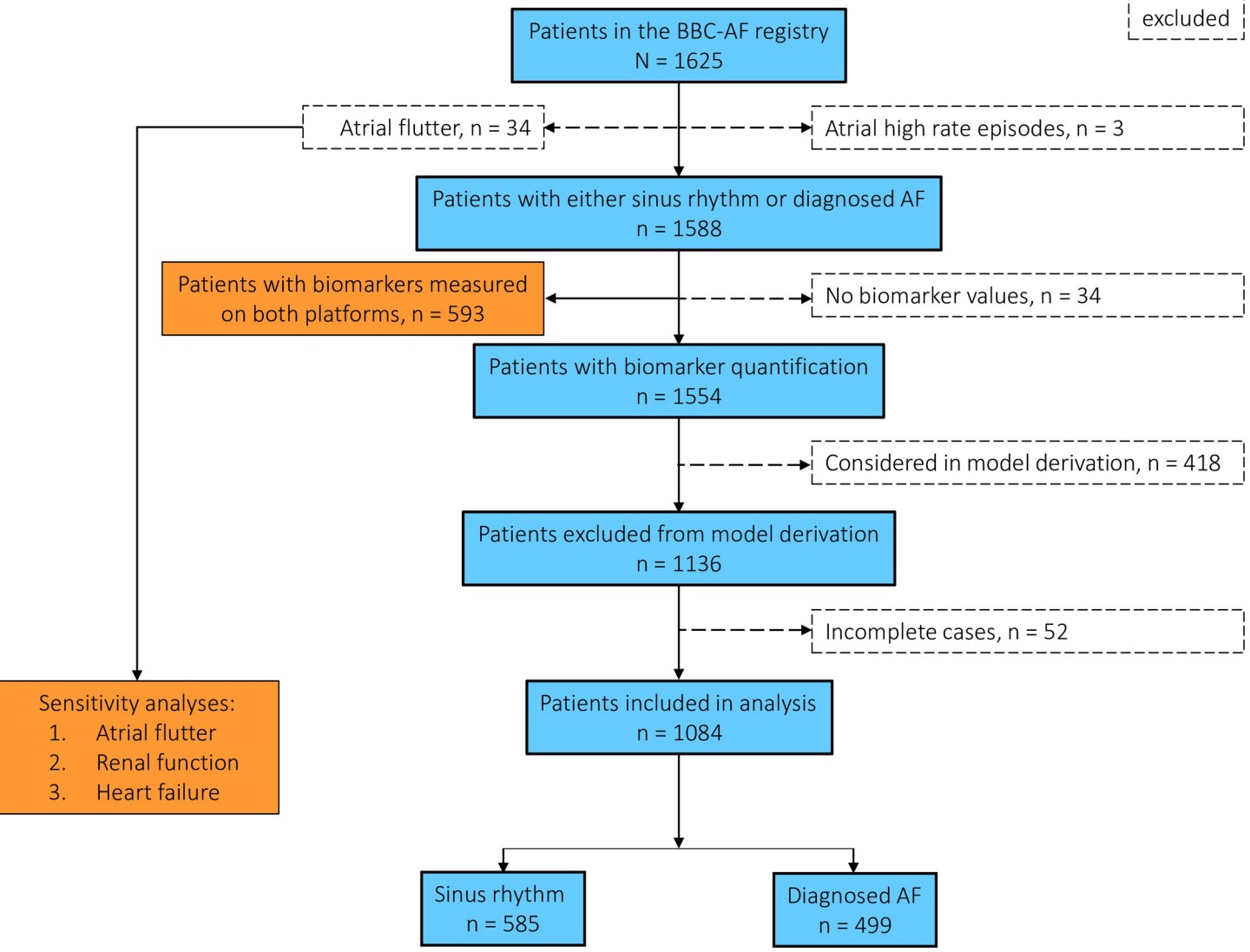

**Fig 1. Flowchart of patients included in the analysis.** A total of 1,084 patients were included in the validation study. Silent or asymptomatic AF patients detected using 7-day ECG monitoring were reassigned to the AF group. Blue boxes indicate the main analysis, orange boxes indicate additional and sensitivity analyses. AF, atrial fibrillation; BBC-AF registry, Birmingham and Black Country Atrial Fibrillation Registry.

sponsored by the University of Birmingham, UK. All patients provided written informed consent.

## Biomarker quantification

Blood samples from all patients were fractionated and stored at −80°C until analysis. Absolute protein concentrations were centrally quantified (Roche Diagnostics, Penzberg, Germany) in EDTA plasma. NT-proBNP was measured with commercial in vitro diagnostic tests (Elecsys NT-proBNP II; Roche Diagnostics, Penzberg, Germany) on a cobas Elecsys analyser (Roche Diagnostics). An immunoassay was developed for detection of FGF23, applying rabbit mono-clonal antibodies screened for specific analyte detection. The assay reached a functional sensitivity of 4 pg/ml precision and within-run coefficient of variation of 1.7%. Like NT-proBNP, FGF23 was quantified using this pre-commercial assay on a cobas 6000 Elecsys platform (Roche Diagnostics). This in vitro diagnostic (IVD) platform is based on electrochemilumines-cence (ECL) technology for heterogeneous immunoassays, allowing up to 170 tests/hour. For both biomarkers, the identical plasma sample per blood draw was measured, blinded to clinical information.

## Statistical analysis

Following the initial discovery of FGF23 and NT-proBNP as markers for AF from a pool of 92 cardiovascular biomarkers [6], this analysis was prospectively planned as part of the extended work plan in the CATCH ME EU-funded project [7] to validate the findings using a high-throughput assay platform that can be implemented in routine clinical care. While the analyses were prospectively planned and agreed on between the authors, we did not write a formal sta-tistical analysis plan.

Baseline characteristics of patients with and without AF were compared. Categorical vari-ables were assessed using $\chi^2$ tests. Continuous variables were compared using independent samples $t$ tests or Mann–Whitney U tests as applicable after determining data normality (Kol-mogorov–Smirnov test and visual inspection of descriptive plots). A 2-tailed $P$ value of $<0.05$ was considered statistically significant. Univariate analyses (adjusted and unadjusted) for study outcome (rhythm status: AF or no AF) were performed. To fit the current data to the published model, the model's beta coefficients and constant were re-estimated to include only biomarkers whose increase indicated higher odds of having AF. This will improve the practi-cality of clinical measurement. The re-estimated model consisted of age, sex, BMI, BNP, and FGF23 (Table A in S1 Text). As the model was derived using BNP, NT-proBNP values were adjusted by a factor of 3 to approximate the established ratio of rule-out cutoffs in acute heart failure [8]. The biomarker data were transformed into the log scale ($\log_2$) to approximate the behaviour of the data used to derive the original model. Furthermore, we compared the perfor-mance of NT-proBNP and FGF23 quantified using Olink and Roche assays, using data from patients who had biomarkers measured on both platforms.

For validation, the re-estimated model was applied to data from the validation cohort. The distributions of demographic characteristics of the validation cohort are very similar to those of the development cohort [6] (Table B in S1 Text). Model performance was evaluated by cal-culating the Brier score, indicating overall performance, and inspecting the area under the receiver operating characteristic (ROC) curve (AUC), the corresponding 95% confidence intervals (CIs), and the elements of a confusion matrix (sensitivity, specificity, positive predic-tive value [PPV], and negative predictive value [NPV]), which were also evaluated at 4 differ-ent cutoff points for probability of outcome. In an additional sensitivity analysis, we also calculated the simple CHARGE-AF model [9,10] for patients in whom all elements of the

11-variable score were available ($n = 1{,}050$) and compared its performance to the biomarker model. We performed sensitivity analyses including patients with atrial flutter, adjusting for renal function, and adjusting the NT-proBNP cutoffs for AF patients with heart failure.

## Results

### NT-proBNP and FGF23

In patients with biomarkers quantified on both Olink and Roche platforms ($n = 593$; Table C in S1 Text), we found that NT-proBNP and FGF23 values correlated highly between platforms (NT-proBNP, $r = 0.867$, $p < 0.001$; FGF23, $r = 0.899$, $p < 0.001$), and variability of the measurements was low (Fig 2).

Of 1,084 patients included in the main analysis, excluding patients used to identify the model [6], 46% ($n = 499$) had diagnosed AF. Baseline characteristics indicated that patients with AF were older and more often had heart failure. The median BMI was in the overweight category for both groups (Table 1). Both biomarkers were elevated in patients with AF (Fig 3A and 3B). We also stratified NT-proBNP by heart failure status and FGF23 by renal function (Fig 3C and 3D). To facilitate comparison of biomarker values with existing studies, we also report raw biomarker values in picograms/millilitre (Table 1) and present a different visualisation of the data (Fig A in S2 Text).

We adjusted the univariate association of NT-proBNP with AF to account for heart failure status, age, sex, and BMI. FGF23 was adjusted by renal function calculated using the Chronic Kidney Disease Epidemiology Collaboration (CKD-EPI) estimated glomerular filtration rate

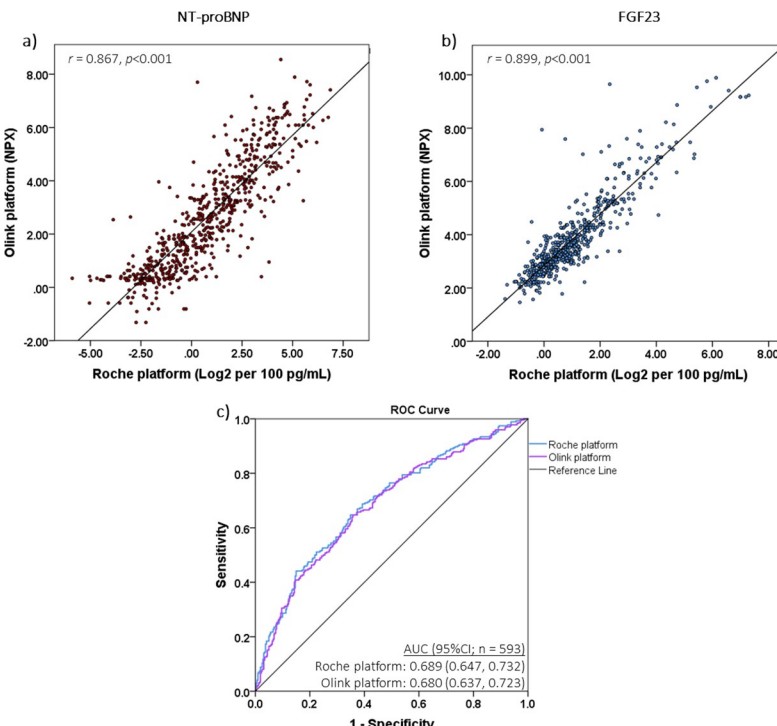

**Fig 2. Comparison between assays.** Values of (a) NT-proBNP and (b) FGF23 measured on the Roche platform highly correlated with those measured on the Olink platform. (c) Model performance calculated using biomarkers quantified on both platforms was identical. AUC, area under the receiver operating characteristic curve; NPX, Normalized Protein eXpression; ROC, receiver operating characteristic.

**Table 1. Descriptive characteristics of patients included in the primary analysis by outcome group.**

| Characteristic | No AF $n = 585$ | AF $n = 499$ | P value | Univariate analysis odds ratio (95% CI) |
|---|---|---|---|---|
| **Age, years** | 67 (57, 75) | 74 (65, 81) | <0.001 | 1.046 (1.035, 1.058) |
| **Sex male** | 344 (59%) | 300 (60%) | 0.66 | 1.056 (0.828, 1.347) |
| **Ethnicity** | | | <0.001 | |
| White | 422 (72%) | 421 (84%) | — | Reference |
| Asian | 99 (17%) | 37 (7%) | — | 0.375 (0.251, 0.559) |
| Black | 64 (11%) | 41 (8%) | — | 0.642 (0.424, 0.972) |
| **BMI, kg/m$^2$** | 28.7 (25.0, 32.7) | 29.0 (25.1, 32.9) | 0.632 | 1.006 (0.987, 1.024) |
| **Renal function** | | | | |
| CKD-EPI | 77.0 (57.0, 92.5) | 69.0 (50.0, 85.0) | <0.001 | 0.986 (0.981, 0.992) |
| MDRD | 74.1 (24.1) | 66.7 (22.4) | <0.001 | 0.989 (0.984, 0.994) |
| Cockcroft–Gault | 80.5 (59.1, 108.0) | 68.7 (51.1, 98.3) | <0.001 | 0.994 (0.991, 0.997) |
| **CKD by stage (CKD-EPI)** | | | <0.001 | |
| I | 177 (30%) | 80 (16%) | — | Reference |
| II | 248 (42%) | 230 (46%) | — | 2.052 (1.491, 2.824) |
| IIIa | 89 (15%) | 93 (19%) | — | 2.312 (1.561, 3.424) |
| IIIb | 44 (8%) | 66 (13%) | — | 3.319 (2.087, 5.278) |
| IV | 22 (4%) | 27 (5%) | — | 2.715 (1.458, 5.057) |
| V | 5 (1%) | 3 (1%) | — | 1.327 (0.310, 5.691) |
| **Diabetes** | 266 (46%) | 126 (25%) | <0.001 | 0.405 (0.313, 0.525) |
| **Stroke/TIA** | 51 (9%) | 48 (10%) | 0.608 | 1.114 (0.737, 1.685) |
| **Coronary artery disease** | 307 (53%) | 122 (24%) | <0.001 | 0.293 (0.226, 0.380) |
| **Hypertension** | 368 (63%) | 254 (51%) | <0.001 | 0.661 (0.480, 0.779) |
| **Heart failure** | 249 (43%) | 268 (54%) | <0.001 | 1.566 (1.231, 1.991) |
| **Admission criteria (inpatient)** | 560 (96%) | 373 (75%) | <0.001 | 0.132 (0.084, 0.207) |
| **Medication** | | | | |
| NOAC | 13 (2%) | 239 (48%) | <0.001 | 40.466 (22.711, 72.030) |
| VKA | 12 (2%) | 122 (24%) | <0.001 | 15.452 (8.421, 28.355) |
| Aspirin | 421 (72%) | 129 (26%) | <0.001 | 0.136 (0.104, 0.178) |
| Antiplatelet agents | 331 (57%) | 94 (19%) | <0.001 | 0.178 (0.135, 0.235) |
| ACE inhibitors | 247 (42%) | 166 (33%) | 0.002 | 0.682 (0.532, 0.874) |
| Angiotensin II receptor blocker | 87 (15%) | 90 (18%) | 0.16 | 1.260 (0.912, 1.739) |
| Beta-blocker | 352 (60%) | 281 (56%) | 0.199 | 0.853 (0.670, 1.087) |
| Diuretic | 171 (29%) | 212 (43%) | <0.001 | 1.788 (1.391, 2.300) |
| Calcium channel antagonist | 134 (23%) | 70 (14%) | <0.001 | 0.549 (0.400, 0.755) |
| Cardiac glycoside | 4 (1%) | 98 (20%) | <0.001 | 35.498 (12.957, 97.252) |
| Aldosterone antagonist | 33 (6%) | 40 (8%) | 0.12 | 1.458 (0.904, 2.349) |
| Antiarrhythmics | 8 (1%) | 36 (7%) | <0.001 | 5.608 (2.582, 12.182) |
| **Biomarkers** | | | | |
| FGF23 (per 100 pg/ml) | 1.55 (1.04, 2.62) | 1.93 (1.30, 4.16) | <0.001 | 1.046 (1.022, 1.070) |
| NT-proBNP (per 100 pg/ml) | 4.25 (1.17, 15.70) | 12.00 (4.19, 30.15) | <0.001 | 1.006 (1.003, 1.009) |
| Log$^2$ FGF23 (per 100 pg/ml) | 0.64 (0.06, 1.39) | 0.95 (0.38, 2.06) | <0.001 | 1.405 (1.275, 1.549) |
| Log$^2$ NT-proBNP (per 100 pg/ml) | 0.56 (2.49) | 1.87 (2.25) | <0.001 | 1.258 (1.193, 1.327) |

Categorical variables are reported as $n$ (%); continuous variables are reported as mean (standard deviation), or median (quartile 1, quartile 3) for non-parametric distributions. The independent samples $t$ test (or Mann–Whitney U test for non-parametric distributions) and $\chi^2$ tests were used to compare characteristics between patients.

ACE, angiotensin-converting enzyme; BMI, body mass index; CKD, chronic kidney disease; CKD-EPI, Chronic Kidney Disease Epidemiology Collaboration; FGF23, fibroblast growth factor 23; MDRD, Modification of Diet in Renal Disease; NOAC, non–vitamin K antagonist oral anticoagulant; NT-proBNP, N-terminal pro-B-type natriuretic peptide; TIA, transient ischemic attack; VKA, vitamin K antagonist.

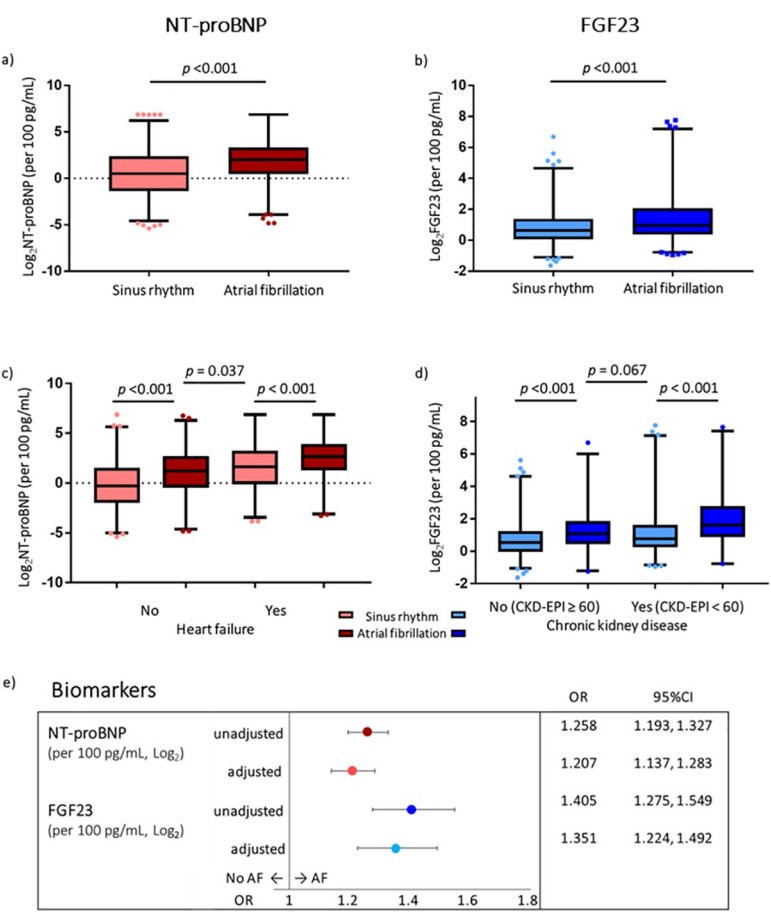

**Fig 3. Description of biomarkers.** Both (a) NT-proBNP and (b) FGF23 are significantly elevated in patients with AF. Stratified biomarker levels indicate that patients with heart failure and impaired renal function have elevated levels of (c) NT-proBNP and (d) FGF23, respectively. Whiskers represent data between the 1st and 99th percentile, with outliers in dots. (e) Adjusted and unadjusted univariate associations of biomarkers (odds ratios, 95% confidence intervals) show significant associations between NT-proBNP, FGF23, and prevalent AF. NT-proBNP levels were adjusted for age, sex, BMI, and heart failure status. FGF23 levels were adjusted for renal function (estimated glomerular filtration rate [ml/min/1.73 m$^2$] calculated using the CKD-EPI equation). AF, atrial fibrillation; CKD-EPI, Chronic Kidney Disease Epidemiology Collaboration; FGF23, fibroblast growth factor 23; NT-proBNP, N-terminal pro-B-type natriuretic peptide; OR, odds ratio.

(eGFR) equation, which takes into account age, sex, and ethnicity, and is normalised by body surface area, using centrally quantified creatinine values obtained in the same plasma sample as FGF23. The univariate associations between the biomarkers and AF outcome remained significant after adjustment (Fig 3E). As a post hoc sensitivity analysis, we also adjusted NT-proBNP and FGF23 by both heart failure and renal function as these diseases may elevate both biomarkers (Fig B in S2 Text).

## Model validation

The model was applied to data from the validation cohort using the re-estimated coefficients, and the AUC was 0.688 (95% CI 0.656, 0.719; Fig 4A), with a Brier score of 0.249. This performance was very similar to the AUC of the derived model, 0.691 (95% CI 0.638, 0.744), with overlapping confidence intervals indicating a remarkably consistent performance of the model in the validation cohort.

a)

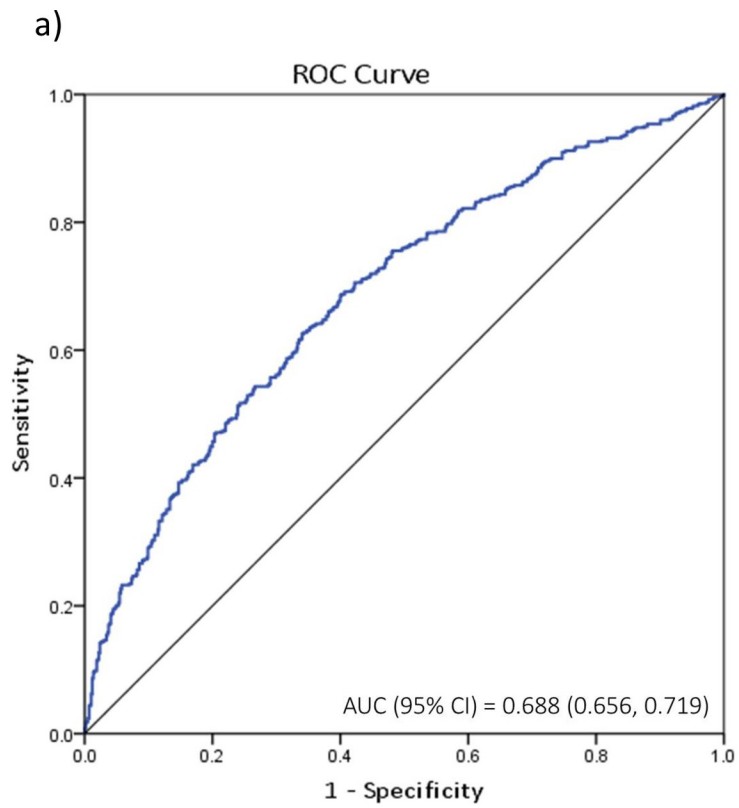

b)

| | Actual | | |
|---|---|---|---|
| Predicted | SR | AF | Total |
| SR | 559 | 404 | 963 |
| AF | 26 | 95 | 121 |
| Total | 585 | 499 | 1084 |

| Parameter | % (95%CI) |
|---|---|
| Sensitivity | 19.04 (15.69, 22.76) |
| Specificity | 95.56 (93.56, 97.08) |
| Positive predictive value | 78.51 (70.66, 84.72) |
| Negative predictive value | 58.05 (56.92, 59.16) |
| Overall Accuracy | 60.33 (57.35, 63.26) |
| Predictions by the biomarker model | n |
| Patients "at risk" | 121 |
| Identified ("at risk", with AF) | 95 |
| Patients "not at risk" | 963 |
| Identified ("not at risk", no AF) | 559 |
| Missed ("not at risk", with AF) | 404 |

**Fig 4. Model validation performance.** (a) The AUC of the validated model (0.688; 95% CI 0.656, 0.719) closely approximated that of the derived model (0.691; 95% CI 0.638, 0.774). (b) The confusion matrix and performance metrics suggest that the model is highly sensitive in being able to identify sinus rhythm patients to "rule out" AF, and has an overall accuracy of 60%. AF, atrial fibrillation; AUC, area under the receiver operating characteristic curve; CI, confidence interval; ROC, receiver operating characteristic; SR, sinus rhythm.

From the confusion matrix, the model had a high specificity (96%) and an overall accuracy of 60% (Fig 4B). When the model was evaluated at different cutoffs, the highest overall predictive accuracy of the tested cutoffs was obtained at 30% (63% accuracy), with a sensitivity of 57% and specificity of 65% (Table 2). The model is presented as a nomogram in Fig 5.

The simple CHARGE-AF model (without echocardiography and ECG variables) [9] was calculated for 1,057 patients ($n = 27$ missing data) and compared to the corresponding biomarker model for the same patients. The 5-variable biomarker model yielded an AUC of 0.692 (95% CI 0.661, 0.724) in these patients, whereas the 11-variable simple CHARGE-AF model had a lower AUC of 0.637 (95% CI 0.604, 0.670). Discrimination measures for this comparison are presented in Fig C in S2 Text.

### Sensitivity analysis

**Atrial flutter.** Atrial flutter is a tachyarrhythmia that often precedes or presents with AF and is managed with similar goals [11]. Therefore, we investigated the performance of the biomarkers in patients with atrial flutter ($n = 29$; $n = 5$ excluded due to incomplete cases). Compared to patients in sinus rhythm, both NT-proBNP ($p = 0.025$) and FGF23 ($p < 0.001$) were significantly elevated in patients with atrial flutter (Table D in S1 Text). When patients with atrial flutter were included in the model validation, the AUC was 0.682 (95% CI 0.651,

Table 2. Model performance metrics at different cutoffs.

| Parameter or prediction | Cutoff | | | |
|---|---|---|---|---|
| | 10% | 20% | 30% | 40% |
| **Parameter, percent (95% CI)** | | | | |
| Sensitivity | 96.99 (95.09, 98.31) | 82.16 (78.52, 85.42) | 56.51 (52.04, 60.91) | 35.07 (30.88, 39.44) |
| Specificity | 7.52 (5.52, 9.97) | 39.32 (35.34, 43.41) | 65.49 (65.49, 73.12) | 86.67 (83.64, 89.32) |
| Positive predictive value | 47.22 (46.53, 47.91) | 53.59 (51.68, 55.50) | 61.17 (57.69, 64.54) | 69.17 (63.89, 74.01) |
| Negative predictive value | 74.58 (62.30, 83.89) | 72.10 (67.61, 76.19) | 65.17 (62.55, 67.70) | 61.01 (59.29, 62.71) |
| Overall accuracy | 48.71 (45.69, 51.73) | 59.04 (56.05, 61.99) | 63.47 (60.52, 66.34) | 62.92 (59.96, 65.80) |
| **Predictions by the biomarker model, *n*** | | | | |
| **Patients "at risk"** | 1,025 | 765 | 461 | 253 |
| Identified ("at risk", with AF) | 484 | 410 | 282 | 175 |
| **Patients "not at risk"** | 59 | 319 | 623 | 831 |
| Identified ("not at risk", no AF) | 44 | 230 | 406 | 507 |
| Missed ("not at risk", with AF) | 15 | 89 | 217 | 324 |

The full model (age, sex, BMI, NT-proBNP, FGF23) was evaluated at 4 cutoff points (probabilities of outcome) that can be used as a decision threshold to identify patients at risk of prevalent atrial fibrillation (AF). The overall accuracy of the model is maximised when a 30% cutoff is used.

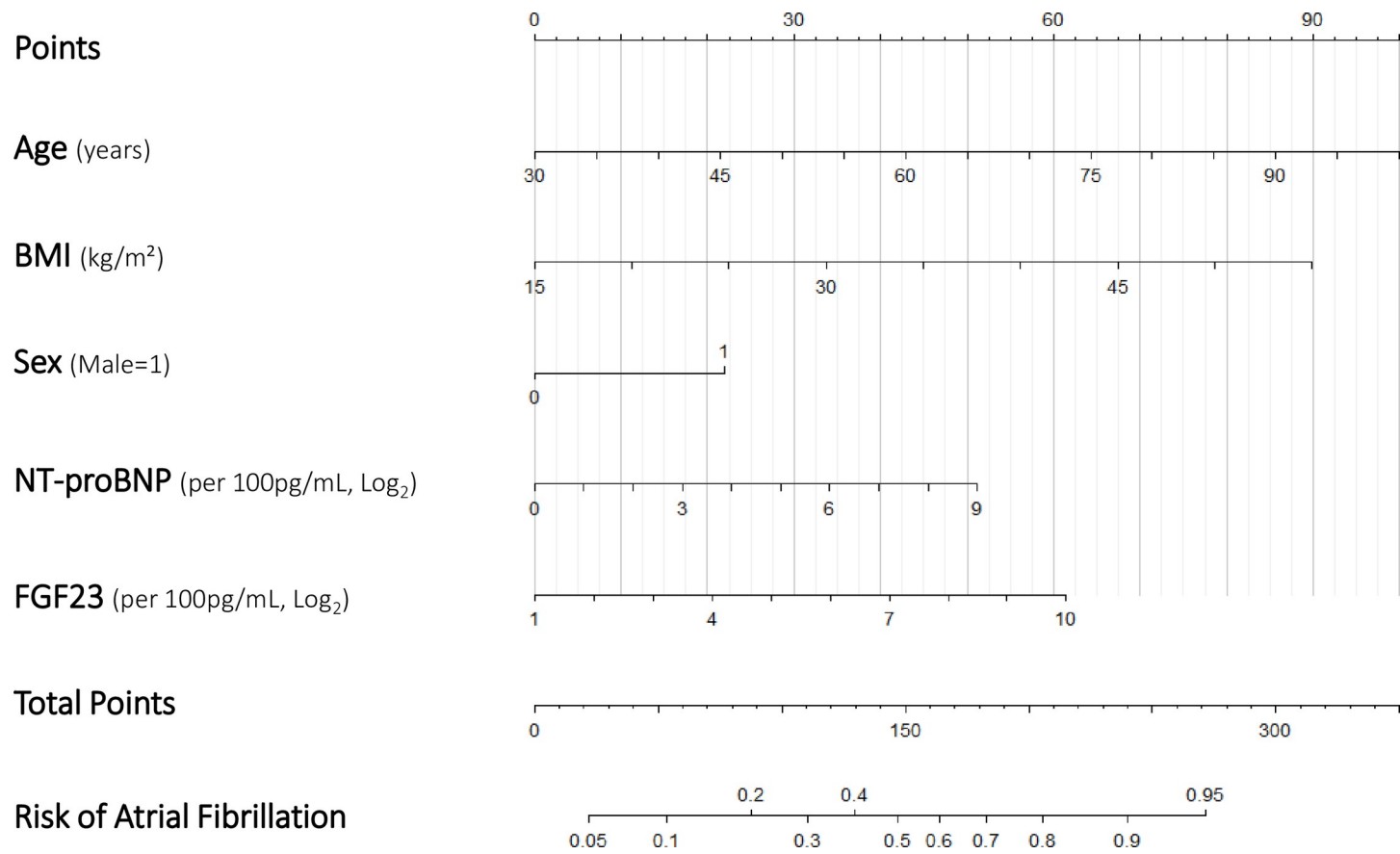

**Fig 5. Nomogram for the proposed biomarker model.** For each patient, points are assigned to each of the 5 variables in the model, which are subsequently totalled to determine the corresponding risk of atrial fibrillation.

**Table 3. Renal function values using different equations.**

| Equation | Mean or median | SD or Q1, Q3 |
|---|---|---|
| MDRD | 71 | 25 |
| CKD-EPI | 73 | 54, 89 |
| Cockcroft–Gault | 75 | 54, 103 |

CKD-EPI, Chronic Kidney Disease Epidemiology Collaboration; MDRD, Modification of Diet in Renal Disease.

0.714), similar to the model without patients with atrial flutter, with overlapping confidence intervals.

**Renal function.** In addition to using the CKD-EPI equation to determine eGFR [12], we performed a sensitivity analysis using the Modification of Diet in Renal Disease (MDRD) equation [13] and the Cockcroft–Gault formula [14] to estimate kidney function. The outcomes of these analyses (Tables 3 and 4) confirm that elevated FGF23 levels remained strongly associated with AF, demonstrating that the relationship between increased circulating FGF23 and AF persists irrespective of renal function (Fig D in S2 Text).

## NT-proBNP cutoffs to detect heart failure in patients with AF

In patients with AF, a higher NT-proBNP threshold value may be more suitable to establish the presence of concomitant heart failure. The reported ROC-optimised cutoffs for ruling in heart failure in the age class of this cohort (50–75 years) in acute settings is 900 pg/ml [15]. In the subcohort of patients with diagnosed AF in the whole BBC-AF Registry ($n = 725$), we assessed the ROC curve of NT-proBNP in predicting the outcome of patients with ($n = 368$) or without ($n = 357$) heart failure. Youden's index was calculated, and the corresponding NT-proBNP level for the maximum value of the index was 1,070 pg/ml, which is approximately

**Table 4. FGF23 levels adjusted by renal function.**

| Analysis | Equation | OR | 95% CI |
|---|---|---|---|
| No correction | **FGF23 (per 100 pg/ml; log$_2$)** | 1.405 | 1.275, 1.549 |
| FGF23 adjusted by renal function measures | **MDRD ($n = 1,084$)** | 1.364 | 1.235, 1.506 |
| | eGFR $> 71$ ml/min/1.73 m$^2$ ($n = 519$) | 1.221 | 1.058, 1.409 |
| | eGFR $\leq 71$ ml/min/1.73 m$^2$ ($n = 565$) | 1.613 | 1.382, 1.882 |
| | **CKD-EPI ($n = 1,084$)** | 1.351 | 1.224, 1.492 |
| | eGFR $> 73$ ml/min/1.73 m$^2$ ($n = 527$) | 1.256 | 1.083, 1.456 |
| | eGFR $\leq 73$ ml/min/1.73 m$^2$ ($n = 557$) | 1.566 | 1.346, 1.821 |
| | **Cockcroft–Gault ($n = 1,084$)** | 1.38 | 1.251, 1.522 |
| | CrCl $> 75$ ml/min ($n = 539$) | 1.217 | 1.060, 1.397 |
| | CrCl $\leq 75$ ml/min ($n = 545$) | 1.549 | 1.333, 1.801 |

Whilst both the MDRD and CKD-EPI equations perform well at eGFR $< 60$ ml/min/1.73 m$^2$, MDRD eGFR is less accurate at eGFR values $\geq 60$ ml/min/1.73 m$^2$, where it tends to underestimate renal function. To avoid potential biases introduced by the renal function equations and subsequently affecting the adjustment, FGF23 levels were corrected by MDRD, CKD-EPI, and creatinine clearance equations in patient subcohorts of renal function values above and below the mean/median for the whole cohort.

CI, confidence interval; CKD-EPI, Chronic Kidney Disease Epidemiology Collaboration; CrCl, creatinine clearance; eGFR, estimated glomerular filtration rate; FGF23, fibroblast growth factor 23; MDRD, Modification of Diet in Renal Disease; OR, odds ratio.

170 pg/ml higher than the recommended ROC-optimised cutoffs for ruling in heart failure in this age category (900 pg/ml) [15].

## Discussion

### Main findings

Using clinical data and centrally quantified biomarkers in >1,000 unselected patients with cardiovascular co-morbidities presenting to hospital, our analysis confirmed that the previously derived model of age, sex, BMI, BNP, and FGF23 identified patients with AF. FGF23 remained elevated in patients with AF after adjusting for renal function, and NT-proBNP remained elevated after adjusting for heart failure. Considering the high probability of silent or asymptomatic AF in the elderly [16], the combination of these 2 biomarkers with simple biometric information might provide a tool to identify patients who need further workup for diagnosing AF (Fig 6). Furthermore, these findings may help to better understand the major drivers of AF in patients.

### NT-proBNP as a marker of cardiac overload in AF

The natriuretic peptide BNP and its N-terminal fragment, NT-proBNP, are produced by cardiomyocytes in response to pressure and volume increase [17]. Natriuretic peptide quantification is clinically used to rule out heart failure [8,18], and elevated levels have been associated

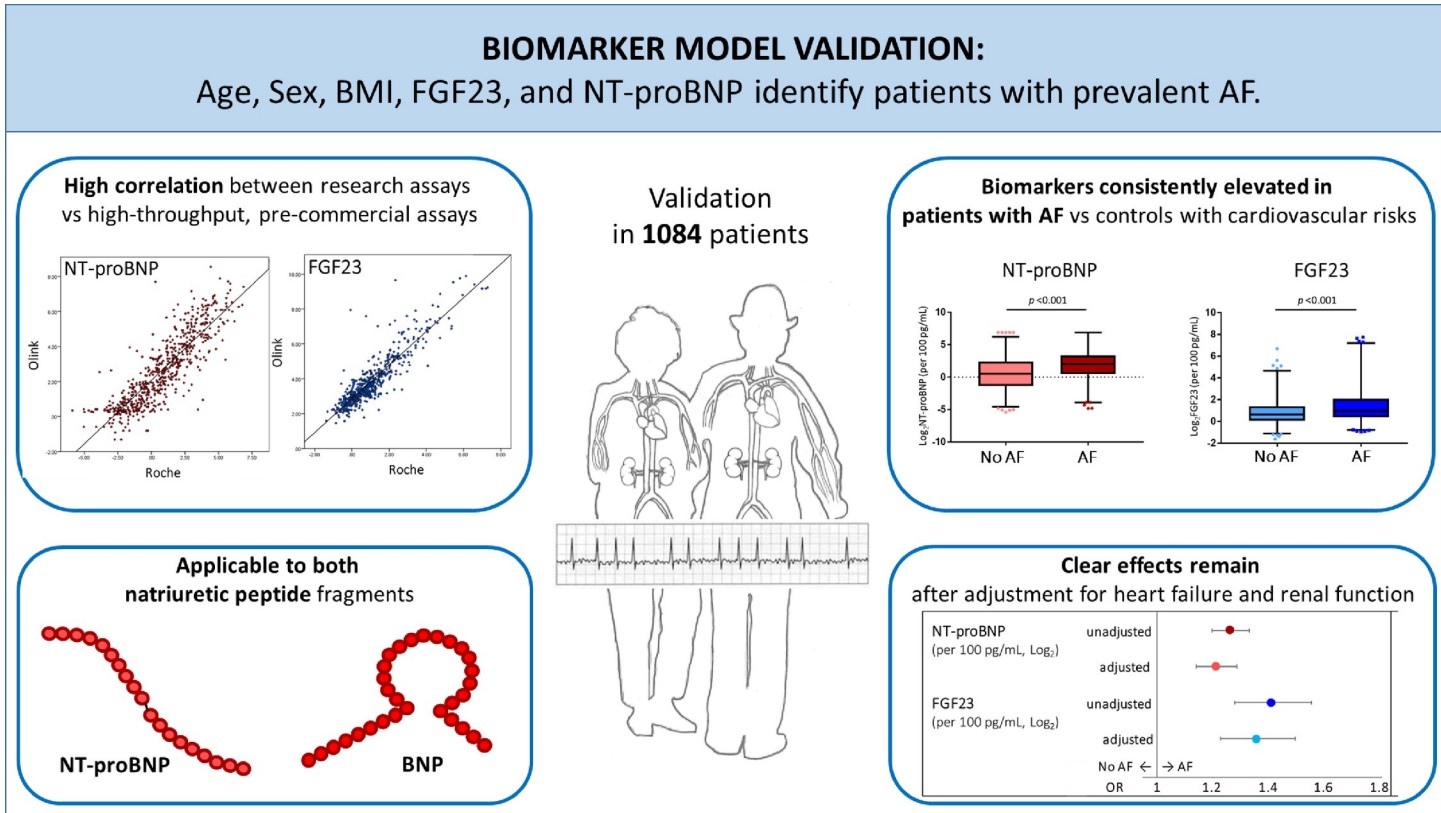

**Fig 6. Validation of a biomarker model for predicting prevalent AF.** Identifying patients with AF to improve screening yield and initiate stroke prevention therapy is an important clinical challenge. We propose the use of a model comprising 3 simple characteristics (age, sex, and BMI) combined with 2 biomarkers (NT-proBNP and FGF23) to identify patients with prevalent AF. AF, atrial fibrillation; OR, odds ratio.

with other cardiovascular conditions. In our prior analysis using a discovery biomarker screening platform, BNP was clearly elevated in patients with AF and emerged as 1 of the 2 strongest out of 92 biomarkers [6]. This finding is highly consistent with prior studies identifying a clear association between AF and elevated natriuretic peptides for predicting imminent AF episodes in patients with paroxysmal AF [19], which has led to testing of natriuretic peptides to guide detection of undiagnosed AF [20], and to inform risk stratification and therapy selection in patients with AF [21,22]. In our validation study, we found that NT-proBNP levels in patients with AF, quantified using standardised, absolute NT-proBNP values, had comparable ORs to the development cohort. The consistency of the presence and magnitude of the associations provides further evidence of the utility of natriuretic peptides for discriminating between patients with and without AF.

Having concomitant AF and heart failure has implications for diagnostic and prognostic performance of NT-proBNP [23]. In our study, clear effects remained after adjusting NT-proBNP for heart failure status, illustrating that both heart failure and AF lead to elevated natriuretic peptides. However, the elevation of NT-proBNP by itself was inadequate to distinguish between heart failure and AF. When we evaluated NT-proBNP cutoffs at 152 pg/ml (5th percentile) and 269 pg/ml (10th percentile) in patients with heart failure and AF ($n$ = 268) in comparison with the recommended cutoff of <125 pg/ml, the recommended cutoff accurately ruled out heart failure in all patients with AF, whereas higher thresholds were less accurate (Table E in S1 Text). This outcome suggests that the current cutoff value for NT-proBNP (<125 pg/ml) is useful in the context of a point-of-care test for further workup as low NT-proBNP values may be able to rule out both AF and heart failure, which have overlapping risk factors and pathophysiologies [24].

## Validation of FGF23 as a marker of AF

The circulating hormone FGF23 is best known as a phosphaturic hormone, with serum levels rising exponentially in chronic kidney disease (CKD), including in the early stages of CKD [25]. Elevated FGF23 is associated with all-cause and cardiovascular mortality [26,27] and with heart failure [26] in patients with or without CKD. In in vitro and in vivo studies, FGF23 has been directly implicated in the development of cardiac hypertrophy [28] and fibrosis [29]. This study validated our prior hypothesis-generating finding that elevated FGF23 increases the odds of prevalent AF. Using a newly developed near-routine, high-throughput, high-sensitivity assay for FGF23 in this validation, we showed that elevated FGF23 in patients with AF is independent of renal function (Tables 3 and 4). Elevated FGF23 has been associated with left ventricular dysfunction and AF in patients without cardiovascular co-morbidities [30], including incident AF after correction for eGFR, albuminuria, and heart failure events [31], and confirmed in patients with CKD [32]. Higher levels of circulating FGF23 were associated with greater odds of incident AF [32]. Coincidentally, our data showed that patients with eGFR and creatinine clearance below the cutoffs (MDRD eGFR $\leq$ 71 ml/min/1.73 m$^2$; CKD-EPI eGFR $\leq$ 73 ml/min/1.73 m$^2$; creatinine clearance $\leq$ 75 ml/min; Tables 3 and 4) have greater odds of having AF, and the cutoffs are around the level at which FGF23 is thought to rise [25].

FGF23 concentrations were similar in patients in sinus rhythm with CKD (CKD-EPI eGFR < 60 ml/min/1.73 m$^2$) and in those with AF and normal renal function (Fig 3D). This observation persisted when applying a higher eGFR threshold or using different eGFR equations (Fig D in S2 Text). FGF23 has been identified to stimulate cardiac remodelling via specific myocardial FGF23-receptor activation. However, this mechanism has been suggested to be independent from alpha-Klotho, which is a co-receptor for FGF23-mediated regulation of mineral metabolism [33]. The failing heart can also be a major source of circulating FGF23,

and patients who experience recurrent AF 6 months after AF ablation were observed to have greater left atrial FGF23 levels and lower soluble Klotho than patients with normal renal function [34,35].

## Potential value for detecting patients with AF

Early detection of AF is crucial for initiation of anticoagulation prior to a first stroke [11] and risk factor management to reduce AF burden [36]. The combination of easily established clinical characteristics and circulating blood biomarkers that can be rapidly quantified presents an important opportunity for point-of-care testing for AF. Both routine and acute presentations to hospital are advantageous moments for patients to be tested and immediately referred for further monitoring where necessary. In these settings, biomarkers can be helpful in identifying at-risk patients [37] who would benefit from prolonged ECG monitoring. Existing models such as CHARGE-AF, require a large number of variables to calculate, some of which require specialist input, e.g., clinical determination of heart failure. As such, a simple model using age, sex, and BMI and a blood sample for biomarker quantification presents an opportunity to accelerate screening efforts, which in our cohort demonstrated better performance than CHARGE-AF. As AF can be driven by different underlying mechanisms [7], it is important for future work to consider multibiomarker models, including novel markers (e.g., cancer antigen 125) [38], in the effort to improve AF detection. In contrast to the research assay platform used in model development, the current high-sensitivity assays validated in this study can be run on routine high-throughput machines already available in many testing labs for immediate, large-scale clinical application.

## Limitations

This validation study was performed in patients recruited using similar criteria as the cohort included in model derivation. While the inclusion criteria were broad and the setting in a general hospital resembles clinical routine in many healthcare systems, thus suggesting that our findings are generalisable, further validation is desirable. Importantly, all patients in this cohort will have a clear indication for oral anticoagulation should they be diagnosed with AF, rendering them a population in whom screening is clinically useful. Our associations, while supporting intriguing mechanistic hypotheses, cannot be used to infer causation. Further studies should evaluate whether elevated NT-proBNP and FGF23 contribute to AF mechanistically or whether they are markers for other disease processes. In addition, although the model was developed for identifying prevalent AF cases, the high specificity of this model suggests that the model might be suited for ruling out non-AF cases. This consideration warrants further analysis for determination of appropriate cutoffs, which is beyond the scope of the current work.

## Conclusions

In this validation study, we demonstrated the consistency of a model combining simple characteristics (age, sex, and BMI) with the biomarkers NT-proBNP and FGF23 to identify patients for further ECG monitoring. The model performed reliably despite potential variability introduced by using different biomarker assays and different natriuretic peptide fragments from those used in the original model. Clear elevations of NT-proBNP and FGF23 remained after rigorous adjustment for known confounders. The outcomes of this study support the use of a multibiomarker model in identifying AF as AF can be driven by different mechanisms. Further validation of this model in different patient cohorts and further basic science investigation into the molecular mechanisms of NT-proBNP and FGF23, especially in the context of AF and co-

morbid heart failure or CKD, would be useful to validate this finding and to understand the drivers of AF linked to elevated FGF23 and NT-proBNP.

## Supporting information

**S1 TRIPOD Checklist.**
(DOCX)

**S1 Text. Tables A to E.**
(DOCX)

**S2 Text. Figs A to D.**
(DOCX)

## Author Contributions

**Conceptualization:** Paulus Kirchhof, Larissa Fabritz.

**Data curation:** Yanish Purmah, Georgiana Neculau, Muhammad Jawad-Ul-Qamar, Kalisha Russell, Ashley Turner, Frantisek Nehaj, Paul Brady.

**Formal analysis:** Winnie Chua, Victor R. Cardoso, Samantha P. Tull, Paul Brady.

**Funding acquisition:** Paulus Kirchhof, Larissa Fabritz.

**Investigation:** Yanish Purmah, Georgiana Neculau, Muhammad Jawad-Ul-Qamar.

**Methodology:** Samantha P. Tull, Peter Kastner, André Ziegler.

**Supervision:** Georgios V. Gkoutos, Paulus Kirchhof, Larissa Fabritz.

**Validation:** Winnie Chua, Victor R. Cardoso.

**Visualization:** Winnie Chua, Victor R. Cardoso.

**Writing – original draft:** Winnie Chua, Jonathan P. Law, Paulus Kirchhof, Larissa Fabritz.

**Writing – review & editing:** Winnie Chua, Jonathan P. Law, Victor R. Cardoso, Yanish Purmah, Georgiana Neculau, Muhammad Jawad-Ul-Qamar, Kalisha Russell, Ashley Turner, Samantha P. Tull, Frantisek Nehaj, Paul Brady, Peter Kastner, André Ziegler, Georgios V. Gkoutos, Davor Pavlovic, Charles J. Ferro, Larissa Fabritz.

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
