## [Editor Report · Decision Letter 0]

11 May 2020

Dear Dr Fabritz, 

Thank you for submitting your manuscript entitled "High-throughput quantification of fibroblast growth factor 23 and N-terminal pro-B-type natriuretic peptide to identify atrial fibrillation." for consideration by PLOS Medicine.

Your manuscript has now been evaluated by the PLOS Medicine editorial staff and I am writing to let you know that we would like to send your submission out for external peer review.

Kind regards,

Helen Howard, for Clare Stone PhD 

Acting Editor-in-Chief

PLOS Medicine 

plosmedicine.org

---

## [Decision Letter · Decision Letter 1]

30 May 2020

Dear Dr. Fabritz,

Thank you very much for submitting your manuscript "High-throughput quantification of fibroblast growth factor 23 and N-terminal pro-B-type natriuretic peptide to identify atrial fibrillation." (PMEDICINE-D-20-01877R1) for consideration at PLOS Medicine. 

[LINK]

In light of these reviews, I am afraid that we will not be able to accept the manuscript for publication in the journal in its current form, but we would like to consider a revised version that addresses the reviewers' and editors' comments. Obviously we cannot make any decision about publication until we have seen the revised manuscript and your response, and we plan to seek re-review by one or more of the reviewers. 

We expect to receive your revised manuscript by Jun 22 2020 11:59PM. Please email us (plosmedicine@plos.org) if you have any questions or concerns.

We look forward to receiving your revised manuscript. 

Sincerely,

Emma Veitch, PhD

PLOS Medicine

On behalf of Clare Stone, PhD, Acting Chief Editor,

PLOS Medicine

plosmedicine.org

*Please revise your title according to PLOS Medicine's style. Your title must be nondeclarative and not a question. It should begin with the main concept/study question if possible followed by the study design in the subtitle after a colon (here, I assumed the data came from a prospective cohort but ": prognostic study" or ": prognostic validation study" could also be used).

*At this stage, we ask that you include a short, non-technical Author Summary of your research to make findings accessible to a wide audience that includes both scientists and non-scientists. The Author Summary should immediately follow the Abstract in your revised manuscript. This text is subject to editorial change and should be distinct from the scientific abstract. Please see our author guidelines for more information: https://journals.plos.org/plosmedicine/s/revising-your-manuscript#loc-author-summary

*In the last sentence of the Abstract Methods and Findings section, please also note briefly any key limitation(s) of the study's methodology.

*It would be good to clarify in the Methods section whether the analytical approach used in the paper corresponded to that set out a priori in a prospective protocol or analysis plan? Please state this (either way) early in the Methods section.

*As noted by one reviewer, it would be good to enhance some aspects of the reporting depth in relation to the analytical approach/general standards for prognostic studies. Two reporting guidelines may be relevant to refer to here (you may wish to choose whichever seems more appropriate, given your study) - TRIPOD (https://www.equator-network.org/reporting-guidelines/tripod-statement/) or STARD (https://www.equator-network.org/reporting-guidelines/stard/). If relevant, you can use the checklist for either guideline to inform reporting changes and also append the completed checklist as supporting information when you resubmit the revised paper. 

Comments from the reviewers:

Reviewer #1: Dear Editors and Authors, 

I read article entitled 'High-throughput quantification of fibroblast growth factor 23 and N-terminal pro-B-type natriuretic peptide to identify atrial fibrillation' with great interest. 

This original investigation concerns important topic: screening for atrial fibrillation (AF) in patients at high thromboembolic risk. 

The abstract incorporates key messages in a concise manner. The general structure of the paper is accurate. 

However, I have some comments to improve the value of the paper:

1. The predictive ability of the model should be compared with CHARGE-AF risk score and BNP or NT-proBNP.

CHARGE-AF risk score includes only clinical risk factors for AF (age, race, height, weight, smoking status, blood pressure, presence of diabetes mellitus, a history of heart failure, use of antihypertensive medication, and a history of myocardial infarction). Please see: EP Europace, Volume 16, Issue 10, October 2014, Pages 1426-1433, https://doi.org/10.1093/europace/euu175. 

This comparison will show whether addition of another biomarker - FGF23 adds significantly to prediction / indication of patients at increased risk of AF.

2. Please propose a nomogram to establish new biomarker-based risk score as e.g. in the ABC (Age, Biomarkers, Clinical history) stroke risk score. Please see e.g. European Heart Journal, Volume 37, Issue 20, 21 May 2016, Pages 1582-1590, https://doi.org/10.1093/eurheartj/ehw054

3. Figure 3 - NT-proBNP levels were adjusted by age, sex, BMI, and heart failure status. It should be adjusted also at least for renal function. Especially in the light of the statement in conclusions ("Clear elevations of NTproBNP and FGF23 remained after rigorous adjustment for known confounders.") please consider adjustment of the data for more potential confounders. Please see e.g.: Clinical Research in Cardiology (2020) 109:331-338. https://doi.org/10.1007/s00392-019-01513-y.

4. The comparison of models in the same patients - the model calculated using Roche quantified biomarkers (AUC 0.689, 95%CI 0.647, 0.732) and the model using Olink quantified biomarkers (AUC 0.680, 95%CI 0.637, 0.723) should be performed using C-statistic. Moreover, assessment of integrated discrimination improvement (IDI) and net reclassification improvement (NRI) should be done to make firm conclusions.

5. In Table S3 data for patients in sinus rhythm should also be provided to make comparison between groups easier.

6. Please introduce abbreviations before the first use, e.g. BMI in the abstract or FDA in introduction. The Authors use two abbreviations for N-terminal pro-B-type natriuretic peptide: NT-proBNP and NTproBNP, only the first one should be used.

7. In the abstract and in the main text - methods section: "≥2 CHA2DS2-VASc stroke risk factors" should be "the CHA2DS2-VASc score of 2 or more' - because as mentioned in another paper originating from the BBC-AF registry (European Heart Journal (2019) 40, 1268-1276) stronger risk factors (age ≥ 75 years old and prior stroke / TIA) could also serve as the only inclusion criteria in the absence of AF.

Reviewer #2: The paper presents a confirmatory study that extends the authors' previously published work. The present study demonstrates that the published model, which is capable of distinguishing between atrial fibrillation and sinus rhythm, generalizes to an entirely new cohort. As additional validation, the authors show that the presence of AF correlates with the levels of BNP and FGF23 even after correcting for heart failure and chronic kidney disease.

Overall the paper is well-written and utilizes standard statistical tests to assess potential associations between univariate measurements and their ability to detect the presence of AF. In general, confirmatory studies are important, and the presented work provides additional evidence for and confidence in the published model. Unfortunately, no attention is given to the new high-throughput platform mentioned in the paper, which artificially lowers the novelty of the presented work.

In the introduction, the authors write "In this study, we developed a novel high-throughput platform for quantifying FGF23, ..." However, the description of this new platform is non-existent. While it is great to see that there is a strong agreement in measurements made with the new and old platforms (Fig 2), it is completely unclear why one would want to use the new platform, if the old one does effectively the same thing. What are some of the features and properties of the new platform (if any) that provide it with an advantage?

Likewise, the discussion states that "Using a newly developed near-routine, high-throughput, high-sensitivity assay for FGF23 in this validation, we showed that elevated FGF23 in patients with AF is independent of renal function." However, without any description of this novel assay, it is again unclear why the same conclusions couldn't be reached with old methods. The authors should include a brief description of the assay and include a link to a standard protocol repository (e.g., https://www.protocols.io/) where more detail is provided.

On a separate note, the paper states that "A two-tailed P value of <0.05 was considered statistically significant." However, given a large number of tests presented in Tables 1 and S2, the authors should include an additional column that shows how p values change after correcting for multiple hypothesis testing.

Minor comments:

- Figure 1: define the meaning of orange and blue boxes

- "The model was fitted with data from the validation cohort..." should read "The model was applied to data from the validation cohort…" (Fitting implies model training.)

Reviewer #3: Within the present manuscript Chua et al. aimed to developed a novel high-throughput platform for quantifying FGF23, used a clinically utilized platform for N-terminal pro-B-type natriuretic peptide (NTproBNP), and quantified these biomarkers in the previously tested patients, as well as in an unselected independent cohort of patients presenting to hospital for validation. The authors concluded that routinely available patient characteristics (age, sex, and BMI) combined with elevated NTproBNP and elevated FGF23, quantified on a high-throughput platform, reliably identify patients with AF.

The scientific question is interesting. However, there are several concerns with respect to the design of the study, data analysis, and presentation of results that need to be considered.

Considering the selection of testes individuals, the authors reported that patients had to present with at CHA2DS2-VASc score of ≥2. It remains unclear why these criteria were applied. The authors need to clarify the rationale behind this definition. The exclusion of low risk (CHA2DS2-VASc score = 0) and intermediate risk (CHA2DS2-VASc score = 1) patients pictures a potentially sicker patient population and does not allow the generalizability of the present results on the general population. 

Moreover, it remains unclear why the authors adjusted for different confounders within the multivariate models for both NTproBNP and FGF23 - since both biomarkers have similar potentially confounding variables.

Considering the purpose of the present analysis, the discrimination and calibration of the presented data needs to be validated by a - in this regard - more relevant NRI and IDI, which is recommended for prognostic biomarkers and clearly illustrated by Cook (Statistical Evaluation of Prognostic versus Diagnostic Models: Beyond the ROC Curve, Clin Chem. 2008 Jan;54(1):17-23.). Since there is an overwhelming increase of biomarker studies in the field of especially cardiovascular disease with varying quality and questionable clinical utility, the evaluation of biomarkers needs to follow stringent rules and regulations. The current investigation does not fulfill all quality criteria recommended by Ahmad et al. (J Am Coll Cardiol HF 2014;2:477-88.) for the evaluation of a prognostic biomarker in cardiovascular disease, including accurate methodological and statistical procedures (Cox regression models, C-statistic, CART-Analysis, NRI and IDI), gain in prognostic information beyond clinically available information and comparison with other biomarkers. The authors need to adjust their analysis and presentation of the results accordingly in order to comply with recommended quality criteria of biomarker research.

[LINK]

---

## [Decision Letter · Decision Letter 2]

12 Aug 2020

Dear Dr. Fabritz,

Thank you very much for re-submitting your manuscript "Quantification of fibroblast growth factor and N-terminal pro-B-type natriuretic peptide to identify patients with atrial fibrillation using a high-throughput platform: validation study." (PMEDICINE-D-20-01877R2) for review by PLOS Medicine.

I have discussed the paper with my colleagues and the academic editor and it was also seen again by xxx reviewers. I am pleased to say that provided the remaining editorial and production issues are dealt with we are planning to accept the paper for publication in the journal.

[LINK]

We look forward to receiving the revised manuscript by Aug 19 2020 11:59PM. 

Sincerely,

Adya Misra, PhD

Senior Editor 

PLOS Medicine

plosmedicine.org

Requests from Editors:

D-20-01877

Title- please add "a" to the study descriptor so the title reads "Quantification of fibroblast growth factor and N-terminal pro-B-type natriuretic peptide to identify patients with atrial fibrillation using a high-throughput platform: a validation study"

Abstract

Please include the details of where the hospital is located and when recruitment took place. 

The abstract needs to follow PLOS Medicine structure using headings (Background, Methods and Findings, Conclusions).

Limitation- please state how the cohort is similar? If you mean demographics, please specify this

Data availability statement- please can you provide a direct contact to the ethics committee as we do not permit authors to be gatekeepers of the underlying data. The same goes for the remaining data, can you please provide a university contact for this. 

Author Summary

Please add bullet points 

Reference brackets should be square brackets throughout please

Please use paragraphs and sections instead of page numbers in the reporting checklist

Line 122 “deeply phenotyped” – is a bit vague. Please remove the ‘deeply’.

Trademark signs should be removed

Comments from Reviewers:

Reviewer #1: Dear Editors and Authors, 

I read substantially improved article entitled 'Quantification of fibroblast growth factor and N-terminal pro-B-type natriuretic peptide to identify patients with atrial fibrillation using a high-throughput platform: validation study' with great interest. 

Thank you for the introduced changes. 

I have only few minor comments now:

1. Please indicate why did you use log2 in terms of biomarkers in the normogram. This is contrary e.g. to a nomogram proposed in the ABC stroke risk score. How could the Readers easily calculate needed values? Or could you simply recalculate the normogram not to include log2 values of biomarkers?

2. Authors summary: please use abbreviations consistently from the beginning of this section. There is no need to write Age or Sex using capital letters.

3. Table S2. The unit of age (I assume years) should be given. Please indicate how is the data presented below the Table. I assume that as median (interquartile range) or number (percentage).

4. In different places of the paper the authors describe the levels of biomarkers per 100 pg/mL. I find it unusual and suggest to describe the level of biomarkers in a standard way.

5. Introduction: please put the FDA abbreviation into the brackets when you introduce it.

6. Supplementary material: please change the title as in the main file of the paper.

Reviewer #3: The authors adequately addressed all raised concerns.

[LINK]

---

## [Editor Report · Decision Letter 3]

21 Dec 2020

Dear Dr. Fabritz,

I am writing concerning your manuscript submitted to PLOS Medicine, entitled “Quantification of fibroblast growth factor 23 and N-terminal pro-B-type natriuretic peptide to identify patients with atrial fibrillation using a high-throughput platform: A validation study..”

We have now completed our final technical checks and have approved your submission for publication. You will shortly receive a letter of formal acceptance from the editor.

Kind regards,

PLOS Medicine